# A Biophysical Approach to the Identification of Novel ApoE Chemical Probes

**DOI:** 10.3390/biom9020048

**Published:** 2019-01-29

**Authors:** Lucas Kraft, Louise C. Serpell, John R. Atack

**Affiliations:** 1Sussex Drug Discovery Centre, School of Life Sciences, University of Sussex, Chichester II, Falmer, Brighton BN1 9QJ, UK; 2Dementia Research Group, Sussex Neuroscience, School of Life Sciences, University of Sussex, Falmer, Brighton BN1 9QJ, UK; l.c.serpell@sussex.ac.uk; 3Medicines Discovery Institute, Cardiff University, Park Place, Cardiff CF10 3AT, UK; atackj@cardiff.ac.uk

**Keywords:** ApoE, Alzheimer’s disease, drug discovery, high-throughput screening, chemical probes, biophysical assays

## Abstract

Alzheimer’s disease (AD) is the most common type of dementia and, after age, the greatest risk factor for developing AD is the allelic variation of apolipoprotein E (ApoE), with homozygote carriers of the ApoE4 allele having an up to 12-fold greater risk of developing AD than noncarriers. Apolipoprotein E exists as three isoforms that differ in only two amino acid sites, ApoE2 (Cys112/Cys158), ApoE3 (Cys112/Arg158), and ApoE4 (Arg112/Arg158). These amino acid substitutions are assumed to alter ApoE structure and function, and be responsible for the detrimental effects of ApoE4 via a mechanism that remains unclear. The hypothesis that a structural difference between ApoE4 and ApoE3 (and ApoE2) is the cause of the ApoE4-associated increased risk for AD forms the basis of a therapeutic approach to modulate ApoE4 structure, and we were therefore interested in screening to identify new chemical probes for ApoE4. In this regard, a high-yield protocol was developed for the expression and purification of recombinant full-length ApoE, and three diverse biophysical screening assays were established and characterized; an optical label-free assay (Corning Epic) for hit identification and microscale thermophoresis (MST) and isothermal titration calorimetry (ITC) as orthogonal assays for hit confirmation. The 707 compounds in the National Institute of Health clinical collection were screened for binding to ApoE4, from which six confirmed hits, as well as one analogue, were identified. Although the compounds did not differentiate between ApoE isoforms, these data nevertheless demonstrate the feasibility of using a biophysical approach to identifying compounds that bind to ApoE and that, with further optimization, might differentiate between isoforms to produce a molecule that selectively alters the function of ApoE4.

## 1. Introduction

Apolipoprotein E (ApoE) is a 299-amino acid glycoprotein that is involved in cholesterol and lipid delivery. Its primarily function is the recognition of specific receptors, such as the low-density lipoprotein receptor (LDLR) that mediates the internalization of lipoprotein particles [1,2]. Apolipoprotein E is highly expressed in the liver, as well as in the brain, where it is mostly secreted by astrocytes [3].

The apolipoprotein E gene (*APOE*) exists as three different polymorphic alleles (ε2, ε3, and ε4) that correspond to variations in the combinations of amino acids at residues 112 and 158 of the protein. While ApoE3 has a cysteine at site 112 and an arginine at position 158, ApoE2 has two cysteines and ApoE4 has two arginines at both sites, respectively [4]. The ε3 allele, and therefore the ε3/ε3 genotype, are most frequent, with an allelic frequency of approximately 78% in the Caucasian population; ε2 and ε4 have frequencies of 8% and 14%, respectively [5]. Studies have shown that the ε4 variant of *APOE* is associated with an increased risk for Alzheimer’s disease (AD) [6,7,8,9], and approximately 40% of AD patients carry at least one ε4 allele [5,10]. The risk of AD increases in a gene dose-dependent fashion: one copy of the ε4 allele increases the risk to develop AD three- to four-fold in comparison to individuals lacking the ε4 allele, whereas two copies of the ε4 allele increase the risk by up to 12-fold [5].

Data suggest that ApoE4 may contribute to AD pathogenesis through both amyloid beta (Aβ)-dependent and Aβ-independent pathways [11], although these processes are still poorly understood. One underlying hypothesis is that a structural difference between ApoE isoforms may confer ApoE4’s detrimental effects in AD. It has been suggested that amino acid substitution in ApoE4 (Arg112 instead of Cys112 compared to ApoE3) alters the protein structure, such that a salt bridge forms between Arg61 and Glu255 which results in an intramolecular interaction between the amino and carboxyl terminal domain of ApoE4; a concept termed domain interaction [12,13,14], with some neuropathological effects of ApoE4 potentially being related to this domain interaction. For example, mice in which the domain interaction is engineered into the mouse ApoE gene [15] show cognitive impairment compared to wild-type matched controls and generally have lower levels of pre- and postsynaptic markers, such as synaptophysin in the hippocampus [16]. Primary cultured astrocytes from these targeted replacement mice have lower levels of ApoE due to the increased activation of unfolded-protein response pathways and degradation of the protein [17]. Similarly, increased neuronal degradation of ApoE4 compared to ApoE3 was demonstrated in mouse neuroblastoma cells, human brain lysates, and in mice expressing ApoE under the control of the neuron-specific enolase promoter [18,19,20]. It has been suggested that an increased rate of proteolytic cleavage of ApoE4 results in fragments that enter the cytosol and interfere with cytoskeletal components, such as tau [20], and may disrupt mitochondrial function [21]. Interestingly, some of these effects can be reversed by disrupting the presumed ApoE4 domain interaction, either by site-directed mutagenesis or by the use of small-molecule “structure correctors” [22].

The modulation of ApoE4 structure to change the protein into an ApoE3-like structure therefore represents a novel therapeutic approach, and tool compounds that directly bind to ApoE4 have been identified [23]. A first set of compounds was described by Ye et al. [24], who screened *in silico* for molecules using the X-ray structure of the ApoE4 amino terminal domain. These small molecules, such as GIND-25 (Azocarmine G), are believed to disrupt the domain interaction [25], restore expression levels of mitochondrial respiratory complexes in cultured brain cortical neurons extracted from ApoE4 transgenic mice [21], as well as restore the processing of ApoE4 through the secretory pathway [26]. A second series of compounds was published a few years later that included molecules with a phthalazinone core structure and were identified using a cell-based fluorescence-resonance energy-transfer (FRET) system [27]. These molecules were also presumed to block ApoE4 domain interaction and were shown to abolish many of its detrimental effects on mitochondrial respiratory function, mitochondrial motility, and neurite outgrowth [27]. Most recently, the detrimental effects of ApoE4 were investigated in cultured neurons derived from human-induced pluripotent stem cells of different *APOE* genotypes and suggest an ApoE4 gain of a toxic function [28]. Many of these toxic effects in cultured neurons could be reduced by treatment with the PH002 phthalazinone [28]. However, phthalazinones were shown to cause significant toxicity in mice [22]. Pyrazoline analogs with similar *in vitro* effects but devoid of *in vivo* toxicity were developed by Merck Research Laboratories and are currently in preclinical trials to establish target engagement and a structure–activity relationship [22,29,30]. Although these molecules have beneficial effects in AD cellular models and are now being investigated in mice for proof-of-concept studies [30], it is not clear as to whether these molecules block ApoE4 domain interaction. A crystal structure of a full-length ApoE4 that could validate the domain interaction is currently missing and thus there remains a lack of direct evidence that these small molecules do indeed alter the structure of ApoE4.

Our aim was to express and purify the different ApoE isoforms and use biophysical assays to enable screening for novel chemical probes. The assays that were optimized were an optical label-free assay (Corning Epic), microscale thermophoresis (MST), and isothermal titration calorimetry (ITC). The 707-compound National Institute of Health (NIH) clinical collection was screened against human full-length recombinant ApoE4, with the label-free assay being used as the primary screen, and MST and ITC being used for confirmation (Figure 1). Taking the screening hits through the secondary assays, we were able to confirm the binding of six drugs, as well as a selected drug analog. All drugs are in clinical use and include a hormone, selective estrogen receptor modulators (SERMs), and nonsteroidal anti-inflammatory drugs (NSAIDs). Although the functional effects of these compounds on ApoE structure and function remain to be elucidated, this study nevertheless demonstrates the feasibility of using biophysical assays to screen for molecules that interact with ApoE4.

## 2. Materials and Methods 

### 2.1. Materials

All materials were purchased from Sigma-Aldrich (Gillingham, UK) or Fisher Scientific (Loughborough, UK) and were the highest purity available. The codon-optimized ApoE4 gene was synthesized and cloned into a pET17b vector at the NdeI/HindIII site by Thermo Fisher Scientific (pET17b_ApoE4). A 6-histidine tag, thioredoxin (TRX), and human rhino virus (HRV) 3C protease cleavage site were upstream of the ApoE4 gene. 

### 2.2. Mutagenesis, Expression, and Purification of ApoE Isoforms and ApoE4 Amino Terminal Domain

The ApoE2 and ApoE3 full-length variants were created by site-directed mutagenesis (QuikChange Lightening site-directed mutagenesis kit, Agilent, Cheshire, UK) using the pET17b_ApoE4 plasmid as template, and primers 5′-ggtgcagatatggaagatgtttgtggtcgtctgg-3′ and 5′-gccgatgatctgcagaaatgtctggcagtttatcag-3′, respectively. Bacterial ApoE proteins were expressed in *E. coli* Rosetta2 (DE3) cells (#71400-3, Merck Millipore, Gillingham, UK) in Luria-Bertani (LB) broth supplemented with 1% (*v/v*) glycerol and 100 µg/mL ampicillin. Protein expression was induced in bacteria grown to an optical density of 600 nm (OD_600_) of 0.7–0.9 with 1 mM isopropyl β-d-1-thiogalactopyranoside (IPTG) for 2 h at 37 °C. Bacteria were then pelleted and resuspended in an ice-cold lysis buffer (50 mM 4-(2-hydroxyethyl)-1-piperazineethanesulfonic acid (HEPES), 240 mM NaCl, 5 mM MgCl_2_, 10 mM imidazole, 1 mM dithiothreitol (DTT), 10% (*v/v*) glycerol, 0.05% (*v/v*) Tween-20, 7 U/mL DNAse I, using one protease inhibitor tablet per 50 mL, pH 8.0). Cell lysis was achieved by sonication and cell debris separated by centrifugation. Proteins were affinity-purified with the Talon (Clontech, Saint-Germain-en-Laye, France) affinity resin. After equilibration of the Talon beads with the soluble component of the cell lysate, the beads were washed with a talon-binding buffer (50 mM HEPES, 240 mM NaCl, 20 mM imidazole, 10% (*v/v*) glycerol, 0.05% (*v/v*) Tween-20, pH 8.0) and protein eluted with talon elution buffer (50 mM HEPES, 240 mM NaCl, 300 mM imidazole, 10% (*v/v*) glycerol, 0.05% (*v/v*) Tween-20, pH 8.0). The elution fractions were then applied on a HiTrap heparin column (GE Healthcare, Little Chalfont, UK) and extensively washed with either a size-exclusion buffer for ApoE3 and ApoE4 (20 mM HEPES, 300 mM NaCl, 10% (*v/v*) glycerol, pH 8.0) or a heparin-binding buffer for ApoE2 (20 mM HEPES, 240 mM NaCl, 10% (*v/v*) glycerol, pH 8.0). After on-column digestion with HRV 3C protease overnight at 4 °C, the cleaved TRX tag was eluted with a size-exclusion or heparin-binding buffer, and ApoE proteins were then eluted by applying a linear salt gradient. Elution fractions were pooled, applied on a HiLoad Superdex 26/600 pg 200 column (GE Healthcare), and all proteins eluted in a size-exclusion buffer. ApoE-containing fractions were concentrated to ~15 mg/mL by centrifugation using VivaSpin 20 concentrators (Sartorius, Surrey, UK, molecular weight cut-off 5000 Da) and protein stored at −80 °C in a size-exclusion buffer (20 mM HEPES, 300 mM NaCl, 10% (*v/v*) glycerol, pH 8.0). All steps and purity of samples were analyzed by sodium dodecyl sulfate polyacrylamide gel electrophoresis (SDS PAGE). The identity of the ApoE isoforms was confirmed by mass spectroscopy.

The ApoE4 amino terminal domain (ApoE4_1-191_) was created by inserting a stop codon into the pET17b_ApoE4 plasmid using primer 5′-gattcgtctgcaggcagaagcagctcaagcccgtct-3′. ApoE4_1-191_ was expressed as for the full-length isoforms. The bacterial pellet was resuspended in ice-cold lysis buffer, sonicated, and cell debris separated by centrifugation. The supernatant was then loaded on a HiFliQ nickel–NTA column (Generon, Slough, UK) and extensively washed with a binding buffer (50 mM HEPES pH 8.0, 300 mM NaCl, 20 mM Imidazole, 10% (*v/v*) glycerol). After on-column digestion with HRV 3C protease overnight at 4 °C, ApoE4_1-191_ was washed out with a binding buffer and the TRX tag eluted, and therefore separated from the amino terminal domain, by applying a linear imidazole gradient. Elution fractions were pooled, applied on a HiLoad Superdex 26/600 pg 75 column (GE Healthcare), and all proteins eluted in a size-exclusion buffer. ApoE_1-191_-containing fractions were concentrated by centrifugation using VivaSpin 20 concentrators (Sartorius, molecular weight cut-off 5000 Da), and protein stored at −80 °C.

### 2.3. Expression and Purification of Human Rhino Virus 3C Protease

The glutathione S-transferase (GST) HRV 3C fusion protein was expressed in *E. coli* Rosetta2 (DE3) cells in Luria-Bertani (LB) broth supplemented with 100 µg/mL ampicillin and 35 µg/mL chloramphenicol. Bacteria were grown to an OD_600_ of 0.6, induced with 0.4 mM IPTG, and protease-expressed overnight at 20 °C. Cells were pelleted and resuspended in an ice-cold lysis buffer (50 mM HEPES pH 8.0, 1000 mM NaCl, 5 mM MgCl_2_, 1 mM DTT, 7 U/mL DNAse I, 1 protease inhibitor tablet per 50 mL), and cell disruption was achieved by sonication. Insoluble components were separated by centrifugation and GST-tagged protease, then affinity-purified using a GSTrap FF column (GE Healthcare). After passing the soluble lysis fractions through the column, the column was extensively washed with a binding buffer (50 mM HEPES pH 8.0, 1000 mM NaCl, 1 mM EDTA, 1 mM DTT) and protease then eluted with 10 mM reduced glutathione. Protease-containing fractions were pooled, concentrated as described above, rebuffered into a storage buffer (50 mM HEPES pH 8.0, 1000 mM NaCl, 1 mM EDTA, 1 mM DTT, 20% (*v/v*) glycerol), and stored at −80 °C.

### 2.4. Activation of Corning Epic Plates and Immobilization of ApoE4

Corning Epic 384-well plates (#5046, Corning, Amsterdam, Netherlands) were activated with 200 mM N-(3-dimethylaminopropyl)-N′-ethylcarbodiimide hydrochloride (EDC) and 50 mM sulfo-N-hydroxysulfosuccinimide (sulfo-NHS) at 26 °C for 30 min. Five microliters of ApoE4 diluted to 300 μg/mL in 20 mM sodium citrate pH 5.6 were dispended into wells and incubated at 26 °C for 1 h. The microplate was subsequently washed three times with phosphate buffered saline (PBS) + 0.1% (*v/v*) Triton X-100 + 2% (*v/v*) dimethyl sulfoxide (DMSO; collectively called PBSTD), and the remaining reactive groups on the biosensor were then blocked by incubation with 50 mM ethanolamine in PBSTD for 5 min at room temperature (RT). The plate was washed three times with PBSTD and, after the final wash, 15 µL PBSTD were added to each well, sealed, and placed into the Epic reader for 1.5 h for thermal equilibration. To measure immobilization levels and/or take the baseline before compound addition after thermal equilibration, 10–15 total points were recorded with 4 × 3 s scans averaged per point.

### 2.5. Suramin-Binding Curve on the Corning Epic

After thermal equilibration of the Corning Epic plate and baseline read (see above), the measurement was paused, the plate removed, and suramin added into the wells (total volume 30 μL) in triplicate at different concentrations, ranging from 0 to 12 mM. The plate was sealed and placed back into the Epic for 30 min. Binding shifts were then measured over 15–20 points in total, with 4 averages per point. Responses were measured as shifts in reflected wavelength and were expressed in picometers (pm).

### 2.6. Screening

The NIH collection (distributed by Evotec, Branford, CT, USA) contains 707 compounds and was screened against ApoE4 using the Corning Epic. Screening was carried out at a compound concentration of 100 μM in two identical runs. Compounds at 10 mM in neat DMSO were diluted to 200 μM in PBS + 0.1% (*v/v*) Triton X-100 (resulting in 2% (*v/v*) DMSO) and then dispensed into the plate (total volume 30 μL) after having read the baseline. Plates were left for 30 min in the machine and binding shifts measured as above. Suramin at 5 mM and 150 μM was used as a positive control and buffer-only was the negative control. Prioritized hits were tested in the form of a concentration response, and were diluted 1:2 in neat DMSO, then transferred into PBS + 0.1% Triton X-100 as above, and added into the wells (total volume 30 μL) in duplicate at different concentration, ranging from 0.2 to 100 μM.

### 2.7. Screening Data Analysis and Hit Selection

Data were extracted with the Epic Analyzer (built 27 February 2015, Corning). Shifts were corrected for background by subtracting the averaged response of the buffer control. Samples with low protein-immobilization levels (<1800 pm) were excluded from data analysis and only plates with a Z′-factor above 0.5 were accepted. The Z′-factor is defined as:(1)$$Z\prime =1-\frac{3\mathrm{SD}\;\mathrm{of}\;\mathrm{sample}+3\mathrm{SD}\;\mathrm{of}\;\mathrm{control}}{\left|\mathrm{mean}\;\mathrm{of}\;\mathrm{sample}-\mathrm{mean}\;\mathrm{of}\;\mathrm{control}\right|}$$
where SD is standard deviation. Hit threshold was set to 5× the median absolute deviation (MAD) of the sample median. For both runs, a median response of 1 pm after buffer correction was determined, as well as a MAD of 3 pm. Thus, a response above 16 pm was set as a hit threshold. Compounds with a negative response and a response >150 pm were not considered a hit.

### 2.8. Microscale Thermophoresis

ApoE4 was labelled in a size-exclusion buffer using the Monolith NT Protein Labeling Kit Red-NHS (#MO-L011, Nanotemper, Munich, Germany) following the manufacturer’s instructions, and the labelling reaction was performed with 20 μM ApoE4 and 40 μM labelling dye. The labelled protein was centrifuged at 16,000× *g* for 5 min at RT before use, and then diluted to 200 nM in 20 mM HEPES, 500 mM NaCl, 10% (*v/v*) glycerol, and 0.1% Triton X-100 (MST buffer). Binding of hits was first tested in eight-point concentration response assays. Compounds at 5 mM in neat DMSO were diluted 1:2 in DMSO and then transferred into MST buffer. Diluted compounds in MST buffer were then combined with equal volumes of 200 nM labelled protein, resulting in a 2% (*v/v*) DMSO working concentration, 100 nM labelled ApoE4, and compound concentrations ranging from 0 to 100 μM. The mixture was pulse-centrifuged and incubated at RT for 5 min. Samples were then centrifuged at 16000× *g* for 5 min and soaked into standard capillaries in duplicates. To achieve full binding curves of compounds in which an effect on thermophoresis or initial fluorescence was observed, 12-point concentration responses were performed from 250 or 500 μM top final concentration in triplicate. For this, compounds were diluted 1:2 in neat DMSO from 12.5 or 25 mM, respectively, transferred into the MST buffer, and combined with equal volumes of 200 nM ApoE4. Microscale thermophoresis measurements were performed on the Monolith NT.115 (Nanotemper). Capillaries were thermally equilibrated at 25 °C for 5 min, and MST traces then collected at 20%–40% light-emitting diode (LED) excitation power and 20%, as well as 40% MST power. Data were analyzed on integrated analysis software (MO Affinity Analyzer version 2.2.4, Nanotemper). A sample denaturation test was performed for the samples that altered the initial fluorescence signal. The compound–ApoE4 mixture was centrifuged for 5 min at 16,000× *g* at RT and then mixed with equal volumes of 2 × denaturing buffer (4% SDS, 40 mM DTT). Samples were denatured at 95 °C for 5 min, soaked into capillaries, and the initial fluorescence was measured. Binding affinities, as presented in Table 1, are the average of 6 independent experiments (±SD).

### 2.9. Isothermal Titration Calorimetry

ApoE isoforms, as well as truncated ApoE4 (ApoE4_1-191_ were dialyzed against PBS + 0.01% (*v/v*) Triton X-100 overnight at 4 °C using 3.5 K Slide-A-LyzerTM dialysis cassettes (ThermoFisher Scientific, Loughborough, UK). ApoE, as well as ApoE4_1-191_ at 300 μM was injected into the cell containing compound at 25–50 μM in PBS + 0.01% (*v/v*) Triton X-100 + 2% (*v/v*) DMSO. Titration experiments were carried out on a Microcal PEAQ-ITC (Malvern Panalytical, Malvern, UK) at 25 °C and 750 rpm, and consisted of an initial (1 μL) injection followed by additional 18 (2 μL) injections. Injection of the protein into PBS + 0.01% (*v/v*) Triton X-100 + 2% (*v/v*) DMSO was used as control experiment and integrated control heats were subtracted from the reaction heats. Data were analyzed on Microcal PEAQ-ITC analysis software (Malvern, version 1.1.0.1262) and data-fitted for one set of sites. Binding affinities as presented in Table 1 are the average of two independent experiments (±SD).

### 2.10. Graphical Representations

Graphical representations of the data were performed using GraphPad Prism version 7.02 (GraphPad Software, Inc, San Diego, CA, USA).

## 3. Results

### 3.1. Expression and Purification of ApoE Isoforms 

We expressed ApoE isoforms in *E. coli* as a TRX fusion protein with a six-histidine tag at the amino terminal domain, and a human rhino virus (HRV) 3C cleavage site between TRX and ApoE. ApoE was purified by a combination of immobilized metal affinity, heparin affinity, and size-exclusion chromatography. All purification steps were under nondenaturing conditions, and the TRX tag was removed by digestion with glutathione *S*-transferase (GST)-tagged HRV 3C protease. The purity of all ApoE isoforms was confirmed to be above 95% (Appendix A). The correct folding of the purified ApoE isoforms was confirmed by circular dichroism (CD) which showed the expected α-helical content for ApoE (Appendix A).

### 3.2. Validation of Biophysical Assays and Suramin as Positive Control

The Corning Epic biochemical label-free assay was used as the primary screen for new ApoE4 tool compounds with MST, as well as ITC being used for hit confirmation (Figure 1). A sodium citrate buffer at pH 5.6 and ApoE4 at a concentration of 300 µg/mL were identified as the optimal conditions to immobilize ApoE4 on Corning Epic biosensors, and used to test the binding of previously described ApoE4 chemical probes: GIND-25 (or Azocarmine G, Sigma-Aldrich, Gillingham, UK) [24], PH002 [27] and EZ-482 [31]. Additionally, we tested suramin for binding since it was reported to decrease the association of ApoE3 to the extracellular matrix of HepG2 cells [32,33,34], and is likely doing this by antagonizing the binding of ApoE to heparan sulfate proteoglycans via the ApoE heparin binding site [35]. Only suramin positively shifted the reflected wavelength, whereas PH002 and EZ-482 had slight negative shifts, which suggests the binding of these compounds to the reference surface of the biosensor. No binding of GIND-25 was detected in the used concentrations (Figure 2). Suramin was therefore used as the positive control and confirmed binding in all biophysical assays (Figure 3).

### 3.3. Screening Identified Fifty-Nine Compounds

The NIH clinical collection (707 compounds) was screened at 100 µM in two separate runs. Suramin was used as a positive control and, at concentrations of 150 µM and 5 mM, produced a mean response measured as shifts in reflected wavelength of 53 ± 11 and 146 ± 17 pm, respectively (Figure 4A). All plates had a Z′ above 0.5 (mean Z′ values of 0.66 ± 0.05 and 0.82 ± 0.04 when using suramin as a positive control at concentrations of 150 µM and 5 mM respectively) and were all included in data analysis (Figure 4B). Shift signals were normally distributed in both runs (data not shown). Wells with low immobilization levels (<1800 pm) were excluded from data analysis and, in total, five wells out of 1920 had to be excluded (Figure 4C).

Out of the 707 screened compounds, 59 met the criterion to be considered a hit (i.e., response greater than fivefold above the median absolute deviation of the sample median; Figure 5A). Of these, 45 compounds were prioritized based on quantitative response and chemical attractiveness, and were tested in concentration response assays on the Corning Epic. Fourteen of these hits gave full or partial binding curves (Figure 5C and Appendix A) and were further evaluated in the secondary assays.

### 3.4. Evaluation of Hits by Microscale Thermophoresis and Isothermal Titration Calorimetry Confirmed Six Drugs 

The hits identified by the initial screening in the Epic assay were next evaluated by MST. Three hits altered ApoE4 thermophoresis in a concentration-dependent fashion, and five altered the measured initial fluorescence (Figure 6). Affinities were determined to be in the micromolar range (Table 1) and some of these drugs, such as clomifene, tamoxifen, and toremifene, share structural features. One confirmed hit, hydroxyzine pamoate, contained pamoic acid as a counter salt, and we were interested to determine whether hydroxyzine itself caused the quenching in initial fluorescence or pamoic acid. The binding of hydroxyzine dihydrochloride and pamoic acid were tested separately and pamoic acid was found to be to be responsible for the response observed with hydroxyzine pamoate (Figure 6D).

Ligand-induced fluorescence changes were tested for specificity and nonspecific effects, such as ligand-induced protein aggregation or loss of material, by denaturing the protein and thus disrupting protein–ligand interaction. In the case of binding-induced quenching, fluorescence intensity in the target and complex sample was equal after denaturation. Similarly, the autofluorescence of ligands has to be excluded as well. Quenching or enhancement of the initial fluorescence by the hits was found to be binding-specific as assessed by the denaturation test and test for autofluorescence (Figure 7).

Finally, the eight hits identified by MST were evaluated by ITC, and six out of eight were confirmed (Figure 8 and Appendix A), with the binding of thioridazine and chlorpromazine to ApoE4 not being confirmed (data not shown). The l-thyroxine analog tafamidis was purchased and also shown to bind to ApoE4 (Figure 8B). Additionally, the binding of all six hits and tafamidis to ApoE2 and ApoE3 was evaluated by ITC, with similar binding to all ApoE isoforms being observed (Appendix A). Interestingly, no binding of the hits to the amino terminal domain of ApoE4 (residues 1–191) could be measured (Appendix A), which suggests the importance of a tertiary/ternary structure, or that the identified drugs bind to the carboxyl terminus of ApoE.

## 4. Discussion

An estimated 47 million people worldwide were living with dementia in 2015, and this figure is projected to double every 20 years to around 130 million by 2050 [36]. Alzheimer’s disease is the most common cause of dementia and accounts for 60% to 80% of all cases [37], but its etiology and pathophysiology remain largely unknown although genetic risk factors seem to work together with environmental and lifestyle factors. The only gene variant that is considered to be an established major risk factor for late-onset AD is the ε4 allele of *APOE* [38]. Hence, many AD patients carry at least one ε4 allele, and studies have shown that the E4 variant additionally decreases the age at onset [5,39]. Consequently, ApoE4 is an interesting potential drug target, and a variety of ApoE4-targeted therapeutic strategies for AD have been suggested [40,41].

The aim of the present study was to validate complementary biophysical assays for the identification of novel ApoE4 chemical probes. Biophysical techniques have become key components of the drug-discovery process for drug targets like ApoE4 that are structurally enabled and are routinely used for hit identification and/or confirmation [42], although these techniques can be limited by the amount of needed protein, sample preparation and throughput. The Corning Epic label-free technology measures a change in the wavelength of refracted light when an analyte binds to a target that is immobilized on a biosensor [43]. The technique is very sensitive and affinities in the nanomolar range can be determined. A microgram of the material is required, and the 384-well plate format allows screening in a high-throughput format. However, buffers must be well-matched for additives and solvents, and nonspecific interactions of compounds with the immobilization medium can result in artificial signals. Furthermore, the covalent binding of the protein to the biosensor may affect protein structure and functionality. Due to the limitations of individual biophysical methods, validation of hits by orthogonal assays, in the present study MST and ITC, are often used to increase confidence in identifying protein–ligand interactions. Microscale thermophoresis monitors the movement of molecules in a temperature gradient, with this movement being tracked by a fluorophore that is covalently coupled to the molecule. The interaction of two molecules (e.g., drug binding to protein) alters this movement and can therefore be used to detect binding [44]. Microscale thermophoresis requires relatively low amounts of material, and thermophoresis curves can give information on unspecific effects such as ligand-induced protein aggregation. Although automated systems are available, the nonautomated capillary-based system used in the present study only allows relatively low throughput (up to 16 samples at a time), and additional validation steps may be required if extrinsic labelling is used. Finally, ITC can provide additional information on thermodynamic parameters and binding stoichiometry. High protein consumption and low throughput limit ITC in hit validation [42]. It is for these various reasons that we decided to use the Corning Epic for hit identification, and MST as well as ITC for hit validation (Figure 1).

Recombinant ApoE4 that was expressed in *E. coli* was used for the identification of novel tool compounds. Unlike plasma- or CSF-derived ApoE, recombinant ApoE does not undergo post-translational modification such as *O*-glycosylation, but has nevertheless been shown to behave in an identical manner to plasma-derived ApoE in terms of LDL-receptor affinity, lipid binding, and stability [45,46]. The correct folding of ApoE isoforms in solution was confirmed by circular dichroism (Appendix A), and the identity of proteins was confirmed by MS that also showed oxidation of some methionine residues (data not shown). Although we assume that methionine oxidation occurred during the sample preparation for MS analysis, it cannot be excluded that residues were already modified during screening, which may have affected affinities and isoform specificity, although this is considered unlikely. Suramin was used as a positive control, although it is neither a specific nor selective ApoE binder, and has been shown to interact with many other proteins [47,48,49]. All our assay buffers contained Triton X-100 detergent to reduce the aggregation of screening compounds and therefore nonspecific binding, as well as to keep ApoE in a physiological relevant conformation (i.e., associated to micelles similar to lipids in lipoproteins). Addition of the detergent was particularly necessary for MST measurements to prevent the binding of ApoE4 to glass capillaries.

Out of the 707 tested compounds, 59 were shown to bind to ApoE4, resulting in a high hit rate of 8%. MST then proved a powerful secondary assay for confirmation studies and reduced the number of hits to eight compounds (1.1% hit rate). Using the MST, we identified pamoic acid to be binding to ApoE4 instead of hydroxyzine, which was the presumed active constituent of hydroxyzine pamoate. Similar observations were made by Zhao et al. [50], who screened for novel agonists of orphan receptor GPR35. In their primary screen, they identified oxantel pamoate as an agonist for GPR35, but subsequent testing with related compounds and pamoic acid revealed pamoate moiety to be responsible for agonist activity. Although pamoic acid does not to bind with high affinity to ApoE4, these results, as well as those of Zhao and colleagues [50], emphasize how the counterion can be the active component and needs to be tested in terms of screening [51]. 

Six out of the eight drugs could be confirmed by ITC. All hits besides pamoic acid are in clinical use and include a hormone (l-thyroxine), three selective estrogen receptor modulators (SERMs; clomifene, tamoxifen, toremifene), and one NSAID (meclofenamic acid). The binding of l-thyroxine to ApoE has previously been reported using a radio-ligand binding assay [52,53] with binding affinities in the micromolar range. Interestingly, the binding of l-thyroxine in the radio-ligand assay was shown to be inhibited by meclofenamic acid [53], which may suggest a shared binding site(s). In contrast to previous studies [52,53], we could not confirm the binding of l-thyroxine or meclofenamic acid to the amino terminal domain but we were able to confirm the binding of the l-thyroxine analog tafamidis to ApoE. Notably, we observed the binding of SERMs to ApoE, which has not previously been described. Selective estrogen receptor modulators are agonists or antagonists of the estrogen receptor, and have been considered as a possible treatment for women with AD. However, it is still a matter of debate if hormone therapy and SERMs have beneficial effects on cognitive function, and clinical studies gave inconsistent results [54].

The hits identified in the present study do not show selectivity for ApoE4 versus ApoE2 and ApoE3, and binding to all ApoE isoforms with similar affinities was observed by ITC. This indicates that ApoE isoforms share structural features, and that compounds bind to conserved binding sites. Affinities are in the micromolar range (Table 1) and it was therefore not possible to accurately determine the stoichiometry for a number of hits. This was most notable with tamoxifen and toremifene, for which it was only possible to confirm binding by ITC but binding affinities could not be measured. Hits were unable to distinguish between isoforms, but did distinguish between full-length and truncated ApoE4, suggesting specific binding to ApoE. In the future, it would be interesting to explore specificity by examining binding to other apolipoproteins such as apolipoprotein AI (ApoA-I). Similarly, as Triton X-100 was added to the assay buffers to keep ApoE in a lipid-like environment, it will also be of interest to test the binding of compounds to lipid-free protein and determine if detergent association affects binding. The exact binding mode of hits to ApoE and their effects on ApoE structure will be addressed in future studies. Despite being nonselective, our compounds might alter ApoE function, and may have beneficial effects, such as on ApoE lipidation and expression, although this remains to be determined. Moreover, they provide chemical starting points for medicinal chemistry efforts for further optimization to attempt to identify isoform-selective binders. 

## 5. Conclusions

In summary, our screening trial successfully identified compounds as novel ApoE chemical probes, and our validated biophysical assays can be used for larger screening initiatives. Although none of the identified hits differentiated between ApoE isoforms, these methods nevertheless provide a novel means of identifying compounds that may be useful in defining the physiological functions of ApoE, and are more generally applicable to investigate other physiological interactions such as ApoE–receptor interaction.

## Figures and Tables

**Figure 1 biomolecules-09-00048-f001:**
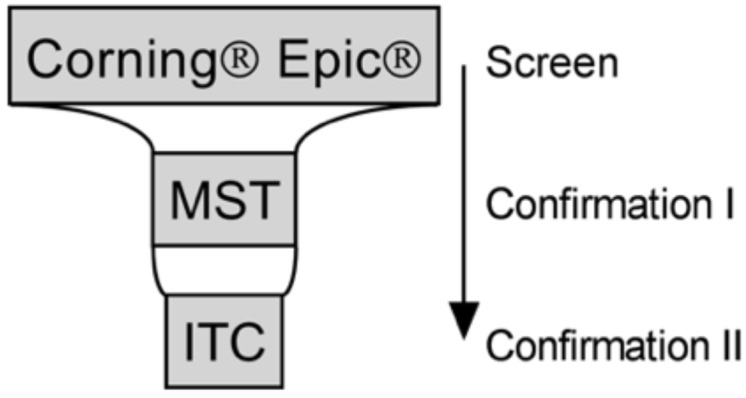
Summary of the screening strategy. Compounds are screened against full-length human apolipoprotein E4 (ApoE4) using Corning Epic and tested by microscale thermophoresis (MST) and isothermal titration calorimetry (ITC). Binding of confirmed hits were then evaluated by ITC against human full-length ApoE2 and ApoE3 to determine selectivity.

**Figure 2 biomolecules-09-00048-f002:**
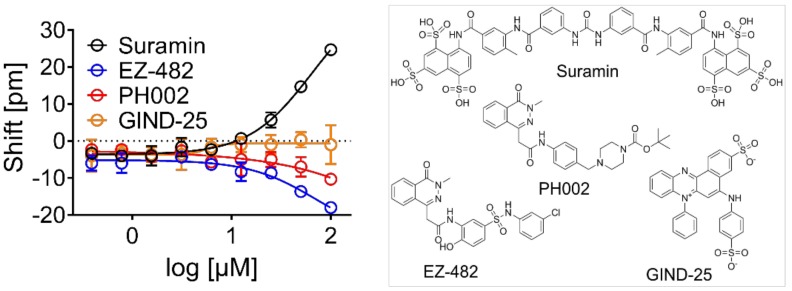
Selection of suramin as positive control. Available ApoE4 modulators, as well as suramin, were tested by concentration response on the Corning Epic against ApoE4. Only suramin positively shifted the wavelength, suggesting binding, whereas negative shifts were observed for PH002 and EZ-482, which may indicate the precipitation of compounds or binding to the reference surface of the biosensor. No binding was detected for GIND-25 at the used concentrations. The data are expressed as the mean of three independent experiments; standard deviation (SD) is represented by the error bars.

**Figure 3 biomolecules-09-00048-f003:**
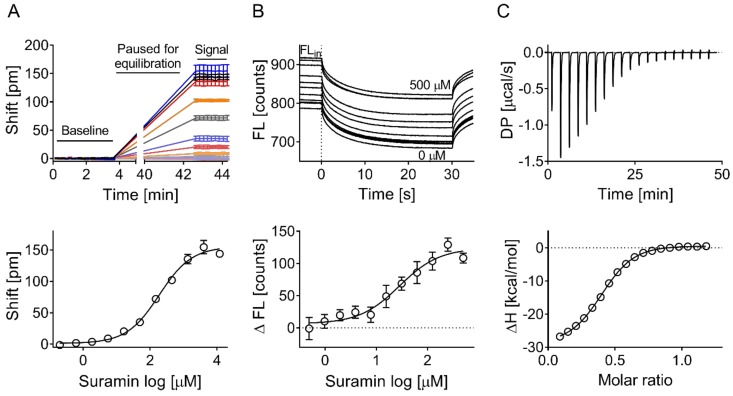
Suramin concentration response across the three biophysical assays. Binding of suramin was confirmed (**A**) on the Corning Epic biochemical assay, (**B**) by microscale thermophoresis, and (**C**) by isothermal titration calorimetry. Mean binding affinities (±SD) of 184 ± 18, 28 ± 8, and 3 ± 0.4 μM were calculated across the three assays, respectively. (**A**) Typically, 10–15 points in total were recorded to measure the baseline before compound or suramin addition. Each point is the average of 4 × 3 s scans. After reading the baseline, the run was paused, the compound added, and the plate returned into the Epic reader for thermal equilibration. The binding signal expressed as shifts in reflected wavelength (picometer) was then measured for 10–15 more points in total. (**B**) Initial fluorescence (Fl_in_) of fluorescently labelled ApoE4 was measured before applying the infrared laser to induce thermophoresis. Suramin increased the initial fluorescence in a concentration-dependent fashion. Data in (**A**) and (**B**) are expressed as mean ± SD of three independent experiments. FL: fluorescence; DP: differential power; ΔH: enthalpy.

**Figure 4 biomolecules-09-00048-f004:**
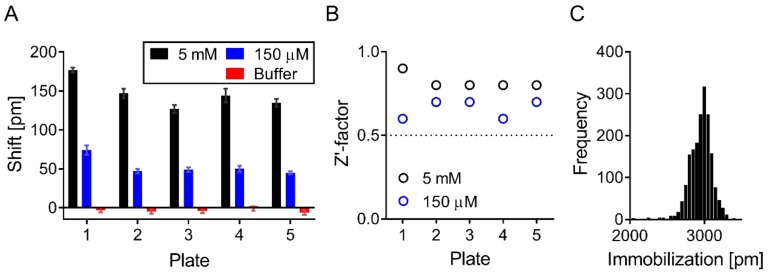
Screening statistics. Suramin at 5 mM and 150 μM was used as positive control, and buffer as negative control during screen. (**A**) Suramin gave a mean shift (± SD) of 146 ± 17 and 53 ± 11 pm at 5 mM and 150 μM, respectively. (**B**) Z′-factors across all plates were >0.5. (**C**) Majority of wells had high ApoE4 immobilization levels, with mean ± SD immobilization levels of 2921 ± 86 pm. Only five biosensors had low ApoE4 immobilization levels (<1800 pm), and were excluded from data analysis.

**Figure 5 biomolecules-09-00048-f005:**
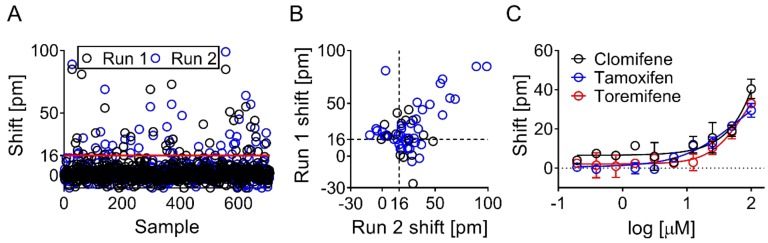
Screening and hit identification on the Corning Epic. (**A**) Hit threshold was set to five times the median absolute deviation (MAD), and a shift above 16 pm was considered a hit. In total, 59 compounds met the criteria to be considered a hit. (**B**) The correlated shifts from both runs are shown. Forty-five of these hits indicated as blue circles were prioritized and tested in 10-point concentration response assays on the Corning Epic, with a top concentration of 100 μM, and 14 compounds showed a concentration-dependent effect, examples of which are shown in panel (**C**). Remaining binding curves are found in Appendix A.

**Figure 6 biomolecules-09-00048-f006:**
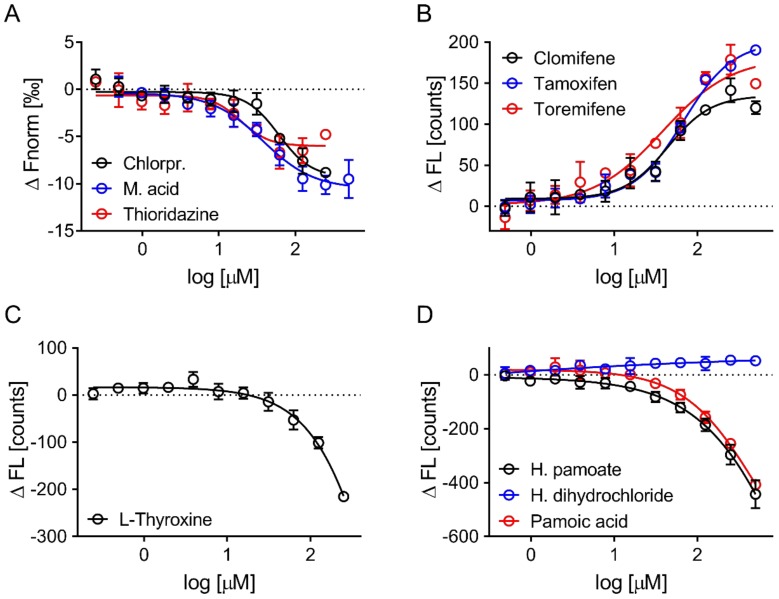
Hit confirmation by microscale thermophoresis. Hits were tested by concentration response from 500 or 250 μM. (**A**) Three out of these eight hits altered thermophoresis without affecting the initial fluorescence (chlorpromazine, meclofenamic acid, thioridazine), (**B**) three increased initial fluorescence (clomifene, tamoxifen, toremifene), and (**C**,**D**) two reduced initial fluorescence (l-thyroxine, hydroxyzine pamoate) in a concentration-dependent fashion. (**D**) Pamoic acid reduced initial fluorescence in a concentration-dependent fashion. Data are expressed as the mean ± SD of three independent experiments.

**Figure 7 biomolecules-09-00048-f007:**
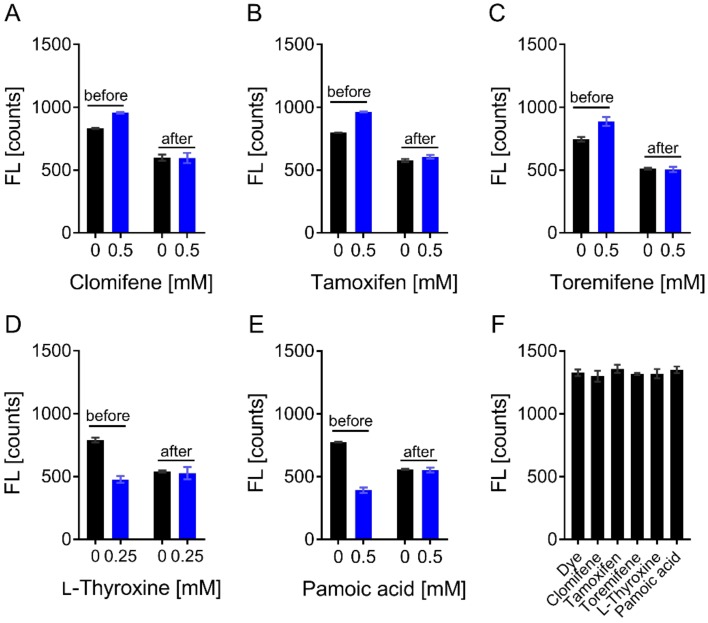
Microscale thermophoresis-denaturation studies and compound autofluorescence. For compounds that altered the initial fluorescence, a denaturation test was performed, as well as tested for autofluorescence. Shown is the initial fluorescence before and after denaturation in the absence and presence of (**A**) clomifene, (**B**) tamoxifen, (**C**) toremifene, (**D**) l-thyroxine and (**E**) pamoic acid. (**F**) Compounds were also tested for autofluorescence in the presence of fluorescence dye without protein. As can be seen, compounds do not influence the fluorescence signal of the dye. Data are expressed as the mean ± SD of three independent experiments.

**Figure 8 biomolecules-09-00048-f008:**
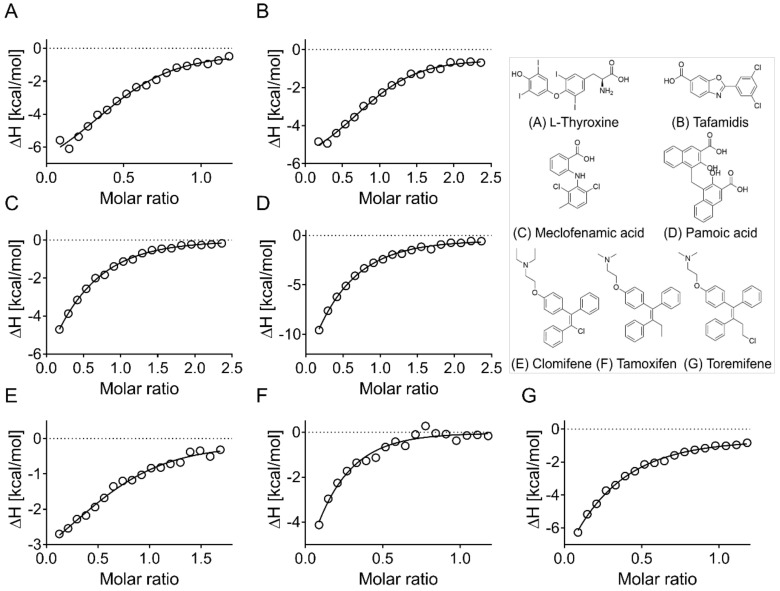
Hit confirmation by isothermal titration calorimetry. All eight hits confirmed by MST were tested by ITC, and six could be confirmed. Binding curves could not be fully resolved of a majority of hits due to binding affinities being in the micromolar range. Binding to ApoE4 was confirmed for (**A**) l-thyroxine, (**B**) its compound analog tafamidis, (**C**) meclofenamic acid, (**D**) pamoic acid, (**E**) clomifene, (**F**) tamoxifen, and (**G**) toremifene. Isotherms of raw titration are shown in Appendix A. Figures show the normalized binding heats, with the solid line representing nonlinear least-square fits using the single-site binding model.

**Table 1 biomolecules-09-00048-t001:** Binding affinities determined by MST and ITC.

	ApoE4		ApoE2	ApoE3
	MST (μM)	ITC (μM)	ITC (μM)	ITC (μM)
l-Thyroxine	*	18 ± 12	33 ± 10	20 ± 3
Tafamidis	-	8 ± 2	6 ± 2	8 ± 1
Meclofenamic acid	51 ± 12	45 ± 13	47 ± 32	47 ± 9
Pamoic acid	*	38 ± 4	55 ± 14	119 ± 50
Clomifene	44 ± 2	15 ± 10	67 ± 36	24 ± 9
Tamoxifen	67 ± 19	*	*	*
Toremifene	44 ± 6	*	*	*

* denotes that binding was confirmed, but binding affinity could not be accurately determined.

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
