# Peer review of "A Biophysical Approach to the Identification of Novel ApoE Chemical Probes"

_biomolecules, 2019, doi:10.3390/biom9020048_

Round 1
Reviewer 1 Report
In their manuscript entitled “A biophysical approach to the identification of novel ApoE chemical probes”, Kraft et al report the use of 3 biophysical assays for screening, hit identification and hit validation of 707 compounds in the NIH Clinical Collection. The Collection is comprised of chemically‐diverse group of drugs with a history of clinical use or having been entered in clinical trials against apoE4, which is considered a risk factor for Alzheimer’s disease.
Using 3 independent assays, the manuscript reports identification of 6 compounds and an analogue of potential interest, in terms of binding to apoE4. The biophysical approaches used are not novel. The study is interesting since it allows the authors to generate a hypothesis that could be tested.
Nevertheless, there are several major fundamental issues with the design and therefore, the interpretation and application of the approaches:
(i) The binding of these compounds is not selective for apoE4, since they also bind apoE3 and apoE2.
(ii) The recombinant proteins were over-expressed in E. coli and likely isolated from inclusion bodies, where it may be misfolded; this would warrant unfolding the protein with a chemical denaturant during or after the isolation procedure. There is no evidence that they employed properly folded and functionally competent apoE proteins. Some evidence that the purified proteins were folded properly (for example by circular dichroism analysis) would help in their assertions.
(iii) It is important to demonstrate that the apoE4 isoform maintains domain-domain interaction (for example, salt bridge formation between R61 and E255) under conditions employed in the assay in the presence of 0.1% TX-100. ApoE4 (and apoE2 and apoE3) is most likely associated with TX-100 micelles following a large conformational change. It is not known at this point if apoE4 retains domain-domain interaction in a lipid (or lipoprotein or lipid micelles)-associated state. Therefore, the premise of testing these compounds, presumably as structure-correctors, is not solid.
(iv) The presence of TX-100 may present a problem in terms non-specific drug binding to the hydrophobic surface presented by the protein-detergent micelles.
(v) Immobilization of apoE for the Corning-Epic optical label- free assay with amine-reactive groups at pH 5.6 may pose a problem since the modification may affect critical functional residues; for example, Lys residues involved in HSPG and LDL receptor binding.
(vi) The concentrations of apoE used in the Corning-Epic optical label- free assay (300 mg/ml) is far above physiological concentrations of apoE in the CNS/CSF (which is ~ 5 mg/ml). What is the physiological relevance of the concentrations used in the study?
(vii) The mean binding affinities calculated from the 3 different assays for Suramin appear to be significantly different with those calculated from Corning-Epic assay being 6- and 30-fold higher than those calculated by MST and ITC, respectively. Is this observation valid?
Incidentally, the reported purification of recombinant human apoE isoforms is not novel as several studies have been published on this topic.
Minor comments:
Several grammatical errors
Author Response
Dear Reviewer,
Thank you for sending us the reviews to our submitted manuscript. We would like to thank you for your well informed and helpful comments. We have considered these carefully and provide an annotated response below. We have revised the manuscript in line with the comments and we believe that this has improved the manuscript substantially. We hope that this paper is now acceptable for publication.
(1) The binding of these compounds is not selective for apoE4, since they also bind apoE3 and apoE2.
It is correct that we could not observe selectivity and all drugs seem to bind with similar affinity to all ApoE isoforms; an observation which is disappointing but not necessarily surprising. Non-selectivity was also shown for the ApoE modulator EZ-482 (a compound analogue of PH002) that interacts with similar affinities with ApoE3 and ApoE4 (Mondal et al., 2016). Although our drugs do not show selectivity, it would be interesting to test in future if they induce structural re-organization of ApoE and if there are differences between ApoE isoforms in this re-organization. Moreover, these compounds could provide the chemical starting points for iterative medicinal chemistry that might be able to dial-in selectivity, although this strategy would clearly benefit from a more detailed understanding of the ApoE-ligand binding modes. These approaches are now mentioned as future directions in the Discussion.
(2) The recombinant proteins were over-expressed in E. coli and likely isolated from inclusion bodies, where it may be misfolded; this would warrant unfolding the protein with a chemical denaturant during or after the isolation procedure. There is no evidence that they employed properly folded and functionally competent apoE proteins. Some evidence that the purified proteins were folded properly (for example by circular dichroism analysis) would help in their assertions.
The recombinant proteins were expressed and purified under non-denaturing conditions as described in the methods section. Furthermore, we have conducted extensive characterisation of the three proteins including circular dichroism (CD), analytical ultracentrifugation (AUC), size exclusion chromatography multi angle laser scattering (SEC MALS) and small angle X-ray scattering (SAXS) and these data are included in a separate paper. We also performed chemical and thermal denaturation studies. We have included an example CD showing the fully folded protein with expected alpha-helical content below.
(3) It is important to demonstrate that the apoE4 isoform maintains domain-domain interaction (for example, salt bridge formation between R61 and E255) under conditions employed in the assay in the presence of 0.1% TX-100. ApoE4 (and apoE2 and apoE3) is most likely associated with TX-100 micelles following a large conformational change. It is not known at this point if apoE4 retains domain-domain interaction in a lipid (or lipoprotein or lipid micelles)-associated state. Therefore, the premise of testing these compounds, presumably as structure-correctors, is not solid.
The evidence for the domain interaction in vivo and with lipids remains unclear. As outlined in the manuscript, although it is an attractive hypothesis, there is no direct proof of ApoE4 domain interaction (i.e. salt bridge R61 - E255) as definitively demonstrated by X-ray crystallography. Other studies hypothesize a structural reorganization and altered charge distribution in the ApoE4 receptor binding region due to the amino acid interchange (Frieden & Garai, 2012); or that ApoE4 may oligomerize differently and as a result may form large, fibril like oligomers (Hatters, Zhong, Rutenber, & Weisgraber, 2006; Kara et al., 2017). With regards to the use of TX-100: ApoE is physiologically present on lipoproteins and being associated to micelles (such as TX-100 micelles) seems physiologically more relevant than being lipid-free. Additionally, ITC was performed at 0.01% TX-100 (instead of 0.1%) which is below the critical micelle concentration of TX-100 (and micelles will not form spontaneously). Binding of hits was therefore tested and confirmed by ITC in the absence of TX-100 micelles.
(4) The presence of TX-100 may present a problem in terms non-specific drug binding to the hydrophobic surface presented by the protein-detergent micelles.
On the contrary, TX-100 was added to the assay buffer to prevent compound aggregation and non-specific binding, as well as to have ApoE in a more physiologically relevant conformation. As outlined in point (3), binding was additionally tested in ITC at 0.01% TX-100, we are therefore not concerned that TX-100 poses a problem in terms of non-specific binding.
(5) Immobilization of apoE for the Corning-Epic optical label- free assay with amine-reactive groups at pH 5.6 may pose a problem since the modification may affect critical functional residues; for example, Lys residues involved in HSPG and LDL receptor binding.
We outline in our discussion the advantages and limitations of the biophysical techniques used in our study. The reviewer makes a valid point and so we have now included a sentence that tethering of the protein to the biosensor may affect structure and functionality of the protein. However, we confirmed our hits by isothermal titration calorimetry (ITC), which is label free and does not require any chemical modification of residues. Hits were therefore tested on functional, unmodified free protein.
(6) The concentrations of apoE used in the Corning-Epic optical label- free assay (300 mg/ml) is far above physiological concentrations of apoE in the CNS/CSF (which is ~ 5 mg/ml). What is the physiological relevance of the concentrations used in the study?
To allow the readings to be above a critical threshold on the Corning® Epic® it is necessary to have good protein immobilization to the biosensor. This was achieved by immobilizing ApoE4 at a concentration of 300 µg/mL (i.e. 0.3 mg/mL, not 300 mg/mL). For the majority of biophysical techniques it is not possible to work at physiological concentrations and increased concentrations are used to ensure sufficient signal to noise. The assays are used to characterise binding of possible compounds which would be explored further in vivo assays following initial in vitro characterisation.
(7) The mean binding affinities calculated from the 3 different assays for Suramin appear to be significantly different with those calculated from Corning-Epic assay being 6- and 30-fold higher than those calculated by MST and ITC, respectively. Is this observation valid?
Given the very different assay methodologies used, it is not surprising that the affinity of Suramin binding to ApoE4 differs according to the method used. With regards to the discrepancy: We determined the Suramin binding affinity on the Corning® Epic® using a one-site binding model. However, ITC indicates two Suramin binding sites per ApoE monomer (i.e. a stoichiometry of 2). Calculating the equilibrium constant using a two-site binding model we determine a KD of 7 ± 1 µM (binding site 1) and 267 ± 43 µM (binding site 2) that indicates high and low affinity binding sites of Suramin to ApoE. The KD of binding site 1 comes very close to values obtained by microscale thermophoresis (MST) and ITC.
Minor comments
We have thoroughly proof-read the manuscript and corrected any obvious grammatical errors.
References used in points (1) and (3)
Frieden, C., & Garai, K. (2012). Structural differences between apoE3 and apoE4 may be useful in developing therapeutic agents for Alzheimer’s disease. Proceedings of the National Academy of Sciences, 109(23), 8913–8918. http://doi.org/10.1073/pnas.1207022109
Hatters, D. M., Zhong, N., Rutenber, E., & Weisgraber, K. H. (2006). Amino-terminal Domain Stability Mediates Apolipoprotein E Aggregation into Neurotoxic Fibrils. Journal of Molecular Biology, 361(5), 932–944. http://doi.org/10.1016/j.jmb.2006.06.080
Kara, E., Marks, J. D., Fan, Z., Klickstein, J. A., Roe, A. D., Krogh, K. A., … Hyman, B. T. (2017). Isoform- and cell type-specific structure of apolipoprotein E lipoparticles as revealed by a novel Forster resonance energy transfer assay. The Journal of Biological Chemistry, 292(36), 14720–14729. http://doi.org/10.1074/jbc.M117.784264
Mondal, T., Wang, H., DeKoster, G. T., Baban, B., Gross, M. L., & Frieden, C. (2016). ApoE: In Vitro Studies of a Small Molecule Effector. Biochemistry, acs.biochem.6b00324. http://doi.org/10.1021/acs.biochem.6b00324
Figure 1. CD of ApoE isoforms. Please refer to uploaded word document.

Reviewer 2 Report
This manuscript describes three biophysical assays (Corning Epic, MST and ITC) used for screening a NIH clinical compound library (707 compound) for the discovery of small molecule binders/chemical probes to recombinant ApoE (E2, E3 and E4) proteins. The described study was partly motivated by the current interest in structural correctors of ApoE4, which have the ability of converting ApoE4 to ApoE3-like structure by disrupting the domain interaction formed by a salt bridge between Arg61 and Glu255 in ApoE4.
The authors were able to demonstrate discovery of small molecules that under conditions used showed interaction with the ApoE proteins. The manuscript discusses well some shortcomings of the current study including recombinant proteins, solubility issues of compounds and non-specific binding to the sensor surface, precipitation of compounds. In my opinion the most significant shortcoming of this study is that the screening assays failed to confirm PH002, a recognized small molecule structure corrector. Nevertheless, the authors successfully identified small molecule binders to ApoE proteins. However, because no modeling and/or functional assays were included in in the described study the biofunctional relevance of the hits/chemical probes is not clear at this point in time (as the authors) correctly state.
Specific Comments:
Line 102: Assuming that the study used the human ApoE4 full length variant; but this should be specified in the method section
Line 59 hits: where are they described and results summarized?
Line 304: 14 hits; where are they described and results summarized?
Line 405: “counterion of salts” seems like a misnomer; pamoate is the counterion (anion) of hydroxyzine (cation)
Author Response
Dear Reviewer,
We thank you for your supportive comments and we have revised the manuscript according to the minor comments below.
Line 102: Assuming that the study used the human ApoE4 full length variant; but this should be specified in the method section
We have specified in the Materials and Methods section if human full length ApoE4 was used, or its truncated version ApoE41-191 or ApoE2/ApoE3. We included now a brief description in Figure 1.
Line 59 hits: where are they described and results summarized?
The 59 hits are summarized in figure 5B as a correlation plot.
Line 304: 14 hits; where are they described and results summarized?
We have now included a supplementary figure showing the binding curves of the 14 hits identified.
Line 405: “counterion of salts” seems like a misnomer; pamoate is the counterion (anion) of hydroxyzine (cation)
We have changed the wording to counterion.

Reviewer 3 Report
Kraft et al describes an approach to identify chemical compounds that binds to the apoE protein. This is performed in three sequential steps involving Corning Epic label-free assay on a compound library (all approved drugs) followed by verification of hits with MST and ITC.
ApoE genotype (E2, E3, or E4) correlates to risk and time of onset of AD. The ApoE isoforms are described to have different structural organizations that may relate to disease risk. The rational for finding apoE binders is thus to find compounds that bind ApoE and "correct" the structure (eg making apoE4 structure more ApoE3-like). In order to perform the analyses the authors establish a bacterial expression system followed by several chromatographic steps for protein purification (similar to earlier described protocols for bacterial production and purification of apolipoproteins). The methods are well described as is also the results section.
Major issues
The authors identify compounds that bind to ApoE but no controls for specific binding is used. Could for example a similar protein such as ApoA-I be used as negative control?
Direct binding of the identified compounds is demonstrated but no information on where in the structure and how the binding affect domain interaction of apoE4 vs the others. This should be analyzed and described.
0.1% Triton is included in the protein preparations. This will likely affect protein structure/function and potentially compound binding. Can the authors comment on this?
Protein identity and sequence of purified ApoE protein should be verified by mass-spectroscopy. MS should also be used to identify the major impurity at about 20kD. The authors should use the knowledge about the identity of this protein to evaluate how it may possibly influence the data and findings.
Authors use truncated 1-191 ApoE and tell about lack of binding (data should be shown) but no gel or QA is shown that verify protein fold etc. Could CD spectroscopy be used for global fold assessment?
Statistical analyses should be used and described eg for data shown in Table 1
Minor
A. Line 42: would be good to describe general frequency in population of apoE4 allele to put "40%" in context
B.Line 134: what is the final concentration? In what buffer? What is the stability of the protein when stored at -80C?
C Line 152: Check sentence (what refers to what)
D. Line 204: what label is used? Specify!
E Line 227 (and elsewhere) ApoE41-191 may be confusing for some readers, I suggest to write apoE4_1-191 or similar.
F. Line 305: examples are shown in panel C. I suggest keep these but to also show teh other as Suppl data.
G. Check references. For example, ref 27 and 31 are the same
H. Fig S1:The text could be more condensed
I. Fig S2: says 0.01% Triton. Is this correct?
Author Response
Dear Reviewer,
Thank you for sending us the reviews to our submitted manuscript. We would like to thank you for your well informed and helpful comments. We have considered these carefully and provide an annotated response below. We have revised the manuscript in line with the comments and we believe that this has improved the manuscript substantially. We hope that this paper is now ready for publication.
(1) The authors identify compounds that bind to ApoE but no controls for specific binding is used. Could for example a similar protein such as ApoA-I be used as negative control?
This is a very good point and a counter-screen against similar proteins such as ApoA-I should be performed in future. We have included this point in the revised discussion. However, we do find differences in binding between truncated and full-length ApoE4 which point to specific binding to the full-length protein (i.e. any nonspecific compound binding would have been observed for the truncated form).
(2) Direct binding of the identified compounds is demonstrated but no information on where in the structure and how the binding affect domain interaction of apoE4 vs the others. This should be analyzed and described.
We have tested binding of compounds by ITC against the ApoE4 amino terminal domain (ApoE41-191) and could not observe binding. As mentioned in our results, this indicates that compounds either bind to the carboxyl terminus or that tertiary/ternary structure is important for drug binding. We did not perform studies on the binding mode (i.e. where compounds bind and how they affect structure) although this would be interesting to evaluate in future studies e.g. by Hydrogen−Deuterium Exchange and Mass Spectroscopy (HDX MS). We have added this future direction in our discussion.
(3) 0.1% Triton is included in the protein preparations. This will likely affect protein structure/function and potentially compound binding. Can the authors comment on this?
We added 0.1% Triton X-100 (TX-100) to our Corning® Epic® and MST assay buffer to avoid aggregation of screening compounds and therefore nonspecific binding. Additionally, 0.1% TX-100 had to be added to the MST assay buffer to prevent binding of ApoE4 to the glass capillaries (initial studies in the absence of TX-100, showed that ApoE4 produced a characteristic “twin-peaks” signal, indicating that the protein sticks to the glass capillaries and influenced thermophoresis). We agree that detergents such as TX-100 are likely to influence ApoE structure and ApoE may be associated to TX-100 micelles. ApoE is physiologically present on lipoproteins and being associated to micelles (such as TX-100 micelles) maybe physiologically more relevant than being lipid-free. Mindful of the potential effects of TX-100, ITC was performed at 0.01% TX-100 (instead of 0.1%) which is below the critical micelle concentration of TX-100. Thus, binding of hits was also tested and confirmed by ITC in the absence of TX-100 micelles.
(4) Protein identity and sequence of purified ApoE protein should be verified by mass-spectroscopy. MS should also be used to identify the major impurity at about 20kD. The authors should use the knowledge about the identity of this protein to evaluate how it may possibly influence the data and findings.
We have confirmed protein identity by mass spectroscopy (MS) and included the results in the Supplementary Data. We added a sentence to our materials and methods. Unfortunately, we did not perform MS on the minor ~22 kDa impurity. ApoE has previously been shown to be prone to proteolysis (Wetterau, Aggerbeck, Rall, & Weisgraber, 1988) and we also observed fragmentation of ApoE over time into ~22 kDa and ~10 kDa products. We therefore assume that the ~22 kDa impurity is an ApoE fragment. However, all studies were performed with freshly prepared protein to ensure as little proteolysis as possible. Additionally, there is only minor amounts present of the ~22 kDa product compared to the full-length protein. We consider this minor impurity not likely to influence binding data.
(5) Authors use truncated 1-191 ApoE and tell about lack of binding (data should be shown) but no gel or QA is shown that verify protein fold etc. Could CD spectroscopy be used for global fold assessment?
We have included now a gel of the ApoE41-191 purification in the Supplementary Data. With regards to folding of ApoE41-191, we crystallized the amino terminal domain and can provide the crystal structure
In terms of folding of full-length ApoE isoforms, we have conducted extensive characterisation of the three proteins including circular dichroism (CD), analytical ultracentrifugation (AUC), size exclusion chromatography multi angle laser scattering (SEC MALS) and small angle X-ray scattering (SAXS) and these data are included in a separate paper. We also performed chemical and thermal denaturation studies. We have included an example CD showing the fully folded protein with expected alpha-helical content below.
(6) Statistical analyses should be used and described eg for data shown in Table 1.
We now have included how binding affinities were calculated in materials and methods. Formal statistical analyses were not conducted since these studies were planned to be descriptive and were not, therefore, powered to demonstrate statistically significant differences between protein and/or compounds.
Minor
A. Line 42: would be good to describe general frequency in population of apoE4 allele to put "40%" in context
We now mention the general frequencies of ApoE alleles in the preceding sentence.
B. Line 134: what is the final concentration? In what buffer? What is the stability of the protein when stored at -80C?
We have added these points to our materials and methods. ApoE seems stable at -80°C; for instance, protein stocks that were stored at -80°C were used to develop the summarizing gel in supplementary figure 1 (gel on the far right).
C. Line 152: Check sentence (what refers to what).
Rewritten.
D. Line 204: what label is used? Specify!
We have added the catalogue number.
E. Line 227 (and elsewhere) ApoE41-191 may be confusing for some readers, I suggest to write apoE4_1-191 or similar.
We have formatted all to ApoE41-191.
F. Line 305: examples are shown in panel C. I suggest keep these but to also show teh other as Suppl data.
We have added a Supplementary Figure showing the other hits.
G. Check references. For example, ref 27 and 31 are the same
We have merged the references.
H. Fig S1:The text could be more condensed.
We have condensed the text.
I. Fig S2: says 0.01% Triton. Is this correct?
That is correct, ITC experiments were performed in the presence of 0.01% TX-100 (not 0.1%).
Reference used in (4)
Wetterau, J. R., Aggerbeck, L. P., Rall, S. C., & Weisgraber, K. H. (1988). Human apolipoprotein E3 in aqueous solution. I. Evidence for two structural domains. The Journal of Biological Chemistry, 263(13), 6240–8. Retrieved from http://www.ncbi.nlm.nih.gov/pubmed/3360781
Figure 1. CD of ApoE isoforms. Please refer to the uploaded word document.

Round 2
Reviewer 3 Report
The authors have significantly improved the manuscript and addressed many of my concerns.
Remaining questions are:
The authors have carried out MS analyses to verify the identity of the purified full-length protein but not the 22kD impurity. Instead the authors discuss that the 22kD is a likely truncated/degraded apoE. The reason for not also determining the identity of 22kD protein is unclear especially as it could easily have been included as an extra sample for the MS.
The MS analyses also reveals significant oxidation of methionine side chains, which is common in oxidative environments (such as air). Such modifications may effect the binding affinities and isoform specificities. At this point, it should at least be described as a potential limitation in the Discussion section.
The authors have performed CD analyses to show that the secondary structures of the three isforms are similar. The data (spectra) should be included as protein QC in the current manuscript (eg as supplemental figure).
As pointed out by the authors in their reply regarding the presence of TritonX100, the majority of plasma apoE proteins are lipid-bound. It would be interesting to compare binding of the compounds to ApoE in the lipid-free and lipid-bound states, and thereby evaluate if the described strategy for compound selection also works for physiological relevant lipoprotein particles. At minimum this should be described as a limitation/future direction in the Discussion.
Author Response
Dear Reviewer,
Thank you for reviewing the revised manuscript. We would like to thank you for your well informed and helpful comments. We have considered these carefully and provide an annotated response below. We have revised the manuscript in line with the comments and we believe that this has improved the manuscript substantially.
(1) The authors have carried out MS analyses to verify the identity of the purified full-length protein but not the 22kD impurity. Instead the authors discuss that the 22kD is a likely truncated/degraded apoE. The reason for not also determining the identity of 22kD protein is unclear especially as it could easily have been included as an extra sample for the MS.
Since the 22 kDa fragment was only a minor component (< 5% of the total purified protein as determined by densitometric scanning of the gel) we considered it very unlikely that it would interfere with our binding data and it is for this reason that we did not conduct an MS analysis of this protein. Our assumption that it is a proteolytic fragment derived from full length ApoE is based on the fact that prolonged incubation of ApoE at room temperature resulted in an increased abundance of this 22 kDa fragment. It is interesting to note that a similar impurity were also observed in previous purification protocols (e.g. Huang et al (1), Figure S1C). The minor impurity is unlikely to influence binding data
Reference:
1. Huang, Y.-W. A., Zhou, B., Wernig, M., and Südhof, T. C. (2017) ApoE2, ApoE3, and ApoE4 Differentially Stimulate APP Transcription and Aβ Secretion. Cell. 168, 427–441.e21
(2) The MS analyses also reveals significant oxidation of methionine side chains, which is common in oxidative environments (such as air). Such modifications may effect the binding affinities and isoform specificities. At this point, it should at least be described as a potential limitation in the Discussion section.
All ApoE proteins were stored in a reducing environment. It is therefore likely that oxidation took place during sample preparation or digestion for MS analysis. However, we have now added this point as a limitation in our discussion.
(3) The authors have performed CD analyses to show that the secondary structures of the three isforms are similar. The data (spectra) should be included as protein QC in the current manuscript (eg as supplemental figure).
CD spectra for the ApoE isoforms is included in the revised manuscript as a supplementary figure.
(4) As pointed out by the authors in their reply regarding the presence of TritonX100, the majority of plasma apoE proteins are lipid-bound. It would be interesting to compare binding of the compounds to ApoE in the lipid-free and lipid-bound states, and thereby evaluate if the described strategy for compound selection also works for physiological relevant lipoprotein particles. At minimum this should be described as a limitation/future direction in the Discussion.
We have added this point as a future direction in our manuscript.
